# Transcytosis subversion by M cell-to-enterocyte spread promotes *Shigella flexneri* and *Listeria monocytogenes* intracellular bacterial dissemination

**Camille Rey**[1], **Yuen-Yan Chang**[1*], **Patricia Latour-Lambert**[1*], **Hugo Varet**[2,3], **Caroline Proux**[2], **Rachel Legendre**[2,3], **Jean-Yves Coppée**[2], **Jost Enninga**[1]*

**1** Institut Pasteur, Dynamics of Host-Pathogen Interactions Unit, Paris, France, and Centre National de le la Recherche Scientifique (CNRS) UMR3691, Paris, France, **2** Institut Pasteur, Transcriptome and Epigenome Platform, Paris, France, **3** Institut Pasteur, Hub Bioinformatique et Biostatistique, Département de Biologie Computationnelle (USR 3756 IP CNRS), Paris, France

☯ These authors contributed equally to this work.

* jost.enninga@pasteur.fr

**Data Availability Statement:** All relevant data are within the manuscript and its Supporting Information files.

## Abstract

Microfold (M) cell host-pathogen interaction studies would benefit from the visual analysis of dynamic cellular and microbial interplays. We adapted a human *in vitro* M cell model to physiological bacterial infections, expression of fluorescent localization reporters and long-term three-dimensional time-lapse microscopy. This approach allows following key steps of M cell infection dynamics at subcellular resolution, from the apical onset to basolateral epithelial dissemination. We focused on the intracellular pathogen *Shigella flexneri*, classically reported to transcytose through M cells to initiate bacillary dysentery in humans, while eliciting poorly protective immune responses. Our workflow was critical to reveal that *S. flexneri* develops a bimodal lifestyle within M cells leading to rapid transcytosis or delayed vacuolar rupture, followed by direct actin motility-based propagation to neighboring enterocytes. Moreover, we show that *Listeria monocytogenes*, another intracellular pathogen sharing a tropism for M cells, disseminates in a similar manner and evades M cell transcytosis completely. We established that actin-based M cell-to-enterocyte spread is the major dissemination pathway for both pathogens and avoids their exposure to basolateral compartments in our system. Our results challenge the notion that intracellular pathogens are readily transcytosed by M cells to inductive immune compartments *in vivo*, providing a potential mechanism for their ability to evade adaptive immunity.

## Author summary

Microfold (M) epithelial cells are important for the onset of infections and induction of immune responses in many mucosal diseases. We extended a human *in vitro* M cell model to apical infections, expression of fluorescent host and microbial reporters and real-time fluorescence microscopy. Focusing on the human intracellular pathogen *S. flexneri*, responsible for bacillary dysentery, this workflow allowed us to uncover that the

**Funding:** CR acknowledges a grant from Danone Research. YYC is supported through a postdoctoral fellowship from the Fondation pour la Recherche Médicale (FRM). JE acknowledges grant support from Institut Pasteur (GPF "MCellHTLV"), the European Union (ERC CoG "EndoSubvert"), and the ANR (Grants "StopBugEntry" and "AutoHostPath"). The DIHP unit is member of the IBEID and MI LabExes. The funders had no role in study design, data collection and analysis, decision to publish, or preparation of the manuscript.

**Competing interests:** The authors have declared that no competing interests exist.

bacterium can subvert the immunological sampling function of M cells by promoting a cytosolic lifestyle and spreading directly to neighboring enterocytes. This mechanism was shared with the etiologic agent of listeriosis, the intracellular pathogen *L. monocytogenes* and allowed both pathogens to avoid exposure to underlying immune compartments. These results may provide a mechanism for the ability of intracellular pathogens to evade adaptive immunity *in vivo*, emphasizing the importance of advanced studies of M cell host-pathogen interactions to understand early steps of mucosal invasion and their consequences on immunity.

## Introduction

Intestinal microfold (M) cells are highly specialized epithelial cells located in the follicle-associated epithelium (FAE), at the interface between the intestinal lumen and the inductive sites of the mucosal immune system. Here, M cells internalize luminal particles and release them to underlying antigen-presenting cells and lymphoid follicles, to prime immunity in the gut associated lymphoid tissue (GALT) [1]. Most major enteric pathogens target M cells as a crucial entry point into the intestinal mucosa, and as such M cells represent the first cell-type involved in the onset of disease and initiation of mucosal immune responses to harmful microbes [2].

The intracellular pathogen *Shigella flexneri* (*S. flexneri)* is reported to cause bacillary dysentery in humans by targeting colonic enterocytes at intestinal crypts, or after translocating via M cells to the basolateral side of the FAE [3–7]. At the FAE, M cell translocation is an important step, as the bacterium is unable to establish its niche in the epithelium by invading adjacent enterocytes from their apical side [8,9]. In the following encounters of *S. flexneri* with target cells, the pathogen expresses a type-three secretion system (T3SS) and virulence factors that drive its invasive behavior [10]. Characteristically, the pathogen enters host cells through a T3SS-dependent trigger mechanism, induces the rupture of the internalization vacuole, and develops a cytosolic lifestyle involving the promotion of actin-based motility (ABM) and cell-to-cell spread via the virulence factor IcsA [10–12]. To date there have been few investigations of *S. flexneri* interactions with M cells and they have remained descriptive, but it has been proposed that *S. flexneri* remain exclusively encased within a vacuole upon M cell transcytosis to basolateral compartments [9,4,5]. It has therefore remained puzzling that despite the immunological sampling, the GALT is unable to mount protective immune responses to secondary infections by this pathogen [13]. In a similar manner, *Listeria monocytogenes* is a human intracellular pathogen thought to translocate the FAE via M cells to promote mucosal invasion, while eliciting poorly protective immune responses [10,14,15]. The interactions of *L. monocytogenes* with M cells have also been succinctly characterized and remain poorly understood. Currently there are no vaccines available against *S. flexneri*, *L. monocytogenes* and many other mucosal-transmitted infectious diseases initiated through M cells [16]. Thus there is a pressing need to elaborate experimental workflows allowing to finely characterize M cell host-pathogen interactions.

Amongst existing approaches to M cell host-pathogen studies, *in vivo* models provide a physiological but complex environment, which is difficult to control. Moreover, as M cells are very rare (they constitute less than 10% of the FAE, which is a rare structure upon itself), animal models yield poorly quantitative data and transient infection events can be difficult to capture [17]. In contrast, *in vitro* experimentation offers controlled levels of complexity and is adaptable to higher throughput techniques, which are well suited for the study of this rare cell-type. Of note, M cells can be induced in small proportions *in vitro*, but not multiplied or

divided [18]. Recently, M cells have been induced in intestinal organoids, but in lack of lymphoepithelial interactions and with the expression of a large panel of differentiated cell types, it is unclear whether this system mimics villous M cells or the FAE [19,20].

Therefore, we decided to use a more established and accessible *in vitro* M cell model mimicking the human FAE, based on the co-culture of human epithelial and lymphoid cell lines [21–23]. In order to track and dissect M cell infection events with increased depth compared to conventional endpoint assays, we adapted this model to physiological apical infections, expression of fluorescent reporters and long-term three-dimensional visualization by time-lapse microscopy.

This workflow was pivotal to reveal that *S. flexneri* develops a bimodal lifestyle within M cells consisting in rapid transcytosis and slower cytosolic invasion via vacuolar rupture. Furthermore, we discovered the ability of cytosolic *S. flexneri* to spread directly from M cells to neighboring cells by IcsA-driven actin-based motility, a process required for the propagation of the epithelial infection. In a similar fashion, we uncovered the ability of *L. monocytogenes* to evade M cell transcytosis completely via ActA and disseminate directly within the FAE by ActA-driven actin-based motility. Our results suggest that intracellular pathogens are able to subvert M cell-mediated immunological sampling, which may have important consequences in our understanding of their ability to evade adaptive immunity *in vivo*.

## Results

### Establishment of a physiological *in vitro* model of *S. flexneri* M cell infections at high spatiotemporal resolution

We investigated the role of M cells during *S. flexneri* infection building on a well-established human *in vitro* model of the intestinal FAE [21–23]. Based on the co-culture of Raji B and Caco-2 cell lines, this model allows the development of a polarized epithelium with physiological proportions (90:10) of enterocytes and M cells, as quantified here by morphological and histochemical analysis. Moreover, particle transcytosis is observed in this system and the expression of relevant M cell differentiation pathways is shown here by single cell transcriptomic analysis (Fig 1A, S1–S3 Figs and S1 Appendix). We extended this model to physiological apical *S. flexneri* infections at low multiplicity of infection (MOI), avoiding widely used infection-enhancement protocols, such as centrifugation or adding poly-Lysine to the bacteria (Fig 1A).

Our physiological infection setup reproduced key features of *S. flexneri* FAE infections observed *in vivo*. Indeed, basolateral *S. flexneri* translocation was noted at 1 hour post-infection (p.i.), under conditions of preserved epithelial barrier integrity (Fig 1B and 1C) [4,5,9]. Moreover, large infection plaques with frequent central erosions of cells were observed at late time-points by confocal microscopy, reminiscent of the epithelial ulcerations observed in shigellosis patients (Fig 1D) [24]. In line with the absence of apical *S. flexneri* invasions, no translocation or infection plaques were observed in monocultures of polarized Caco-2 enterocytes (Fig 1B–1D).

To explore in more detail the dynamics between invading bacteria and M cells at the subcellular level, we developed a protocol for live acquisitions of infected co-culture monolayers, which has no observed incidence on the expression and antigen-sampling function of the M cells. This new approach allowed us to perform three-dimensional time-lapse imaging of *in vitro* FAEs at high spatiotemporal resolution up to 21 hours. We stained the co-cultures with the membrane dye FM4-64, identified M cells with fluorescent WGA, and used *S. flexneri* expressing dsRed for efficient spotting of the infection sites (Fig 1A, S4 and S5 Figs). Of note, this approach required the use of *S. flexneri* expressing the adhesin AfaI to prevent apical

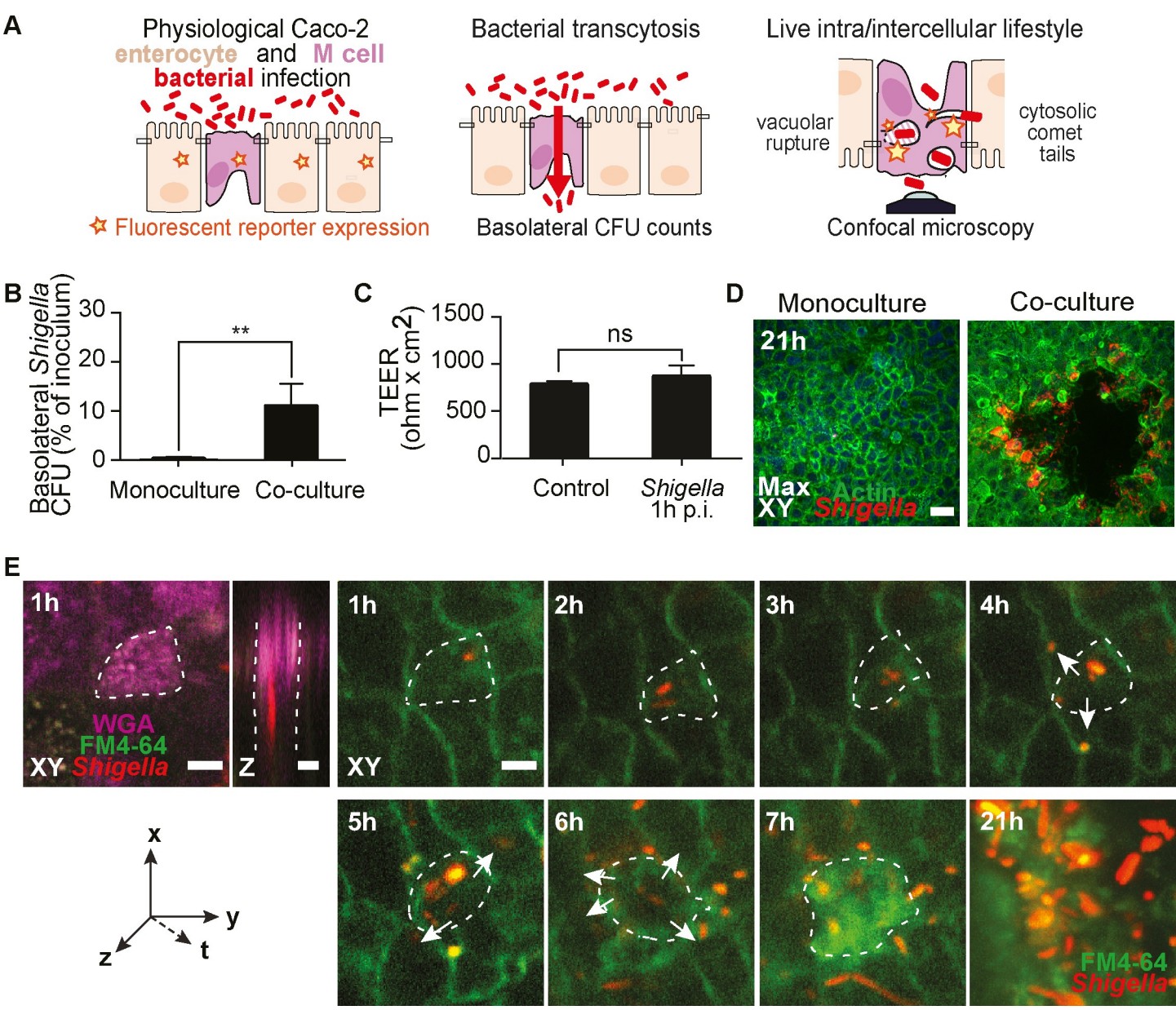

**Fig 1. Establishment of a physiological *in vitro* model of *S. flexneri* M cell infections at high spatiotemporal resolution.** (A) Scheme of the FAE model and its applications with genetically encoded fluorescent reporters, translocation assays and confocal imaging. (B) *S. flexneri* translocates exclusively through M cell containing co-cultures at 1h post-infection (p.i.) and not through polarized monocultures, ($n = 5$, *P <0.01). (C) Co-culture barrier integrity, monitored by transepithelial electrical resistance (TEER) measurements in non-infected (control) and infected monolayers, is preserved at 1h p.i. ($n = 5$). (D) *S. flexneri* infection plaques form exclusively within M cell containing co-cultures (right panel) and not in polarized monocultures (left panel) at 21h p.i. ($n = 5$). Scale bar, 50 μm (E) Representative time-lapse of *S. flexneri* targeting a WGA-positive M cell at 1h p.i. before spreading to neighboring cells from 4 to 6h p.i. (arrows) and generating an infection plaque at 21h p.i.. Scale bar, 20 μm.

detachment of the bacteria from the monolayers during handling for imaging (detailed in Materials and Methods). The time-lapses confirmed the exclusive apical targeting of M cells by *S. flexneri* (varying from 0.05 to 15% of the M cell population) at early time-points and recapitulated the polarized course of infection with the formation of infection plaques at late time-points in the co-cultures (Fig 1E).

## *S. flexneri* follows a bimodal itinerary within M cells

The intracellular lifestyle of *S. flexneri* is well understood in enterocytes, and involves a T3SS-dependent trigger entry into host cells [10], the early rupture of the internalization vacuole [11], and the development of a cytosolic lifestyle involving the promotion of actin-based motility (ABM) and cell-to-cell spread via the virulence factor IcsA [12]. Within M cells however, the general consensus is that *S. flexneri* is internalized exclusively within a vacuole for transcytosis to basolateral lymphoid follicles [5,6].

To test this theory, we performed a translocation time-course assay. *S. flexneri* transcytosis took place early and rapidly, with ~90% of the translocating bacteria reaching the basolateral side of the co-cultures within 15 min p.i. (Fig 2A). Moreover, we confirmed that transcytosis is an invasive process, using an *ΔmxiD* mutant expressing a non-functional T3SS [4,5,25]. In contrast, we observed with a non-motile *ΔicsA* mutant that cytosolic ABM was not required (Fig 2B). These results suggested that *S. flexneri* indeed undertakes a direct route for transcytosis, most likely encapsulated within a vacuole.

To monitor the subcellular localization of *S. flexneri* within co-culture M cells in real time, we generated a stable Caco-2 cell line expressing the vacuolar rupture reporter galectin-3-eGFP, established by Paz et al. [26]. We were not able to capture cytosolic access by time-lapse microscopy during the rapid early transcytosis of *S. flexneri*. Instead, we observed extended events of apical bacterial adhesion to WGA-positive M cells after the occurrence of the majority of transcytosis events, starting from the beginning of our imaging acquisition sessions at around 20 min p.i. (Fig 2C, S1 Movie). These bacteria were often engulfed in apical projections of membranes, visible by the membrane dye FM4-64, and by WGA, similar to membrane ruffles formed during *S. flexneri* enterocyte entry [10]. Furthermore, these interactions were followed by late events of vacuolar rupture, compared to enterocytes [11,27], starting from 74 ± 33 min p.i. This roughly coincided with the onset of intracellular replication of the bacteria, starting from 77 ± 27 min p.i (Fig 2C and 2D, S1 Movie). Hence, we detected the development of a cytosolic population of *S. flexneri* within M cells, thus revealing a bimodal lifestyle consisting in rapid transcytosis to basolateral compartments or slower cytosolic invasion.

## *S. flexneri* spreads directly from the M cell cytosol to neighboring enterocytes via IcsA

It has been speculated that *S. flexneri* infections disseminate to the FAE from basolateral compartments, after transcytosis through M cells [6].

To test this model, we blocked *S. flexneri* propagation from M cells applying gentamicin basolaterally, and using a *ΔicsA* mutant to prevent cytosolic ABM-mediated cell-to-cell spread. Under these conditions, *S. flexneri* infection remained restricted to small foci of infection at late timepoints, in contrast to the large infection plaques observed under conditions where ABM was not impaired. This observation prompted us to quantify the contribution of the two routes- basolateral invasion vs cytosolic ABM- for *S. flexneri* dissemination. Although basolateral bacteria enhanced *S. flexneri* propagation in the FAE, we observed that they were not involved in dissemination from M cells, as their eviction still led to larger areas of infection than those observed in conditions of arrested dissemination. In contrast, the *ΔicsA* mutant failed to spread beyond the initial infection focus, indicating that ABM is required for bacterial dissemination from M cells to the FAE (Fig 3A and 3B, S6A Fig).

To confirm the promotion of ABM by *S. flexneri* within M cell cytosols by time-lapse microscopy, we generated a stable Caco-2 reporter cell line expressing villin-eYFP, established by Revenu et al. [28], for actin comet tail visualization. Starting from 90 ± 46 min p.i., bacterial

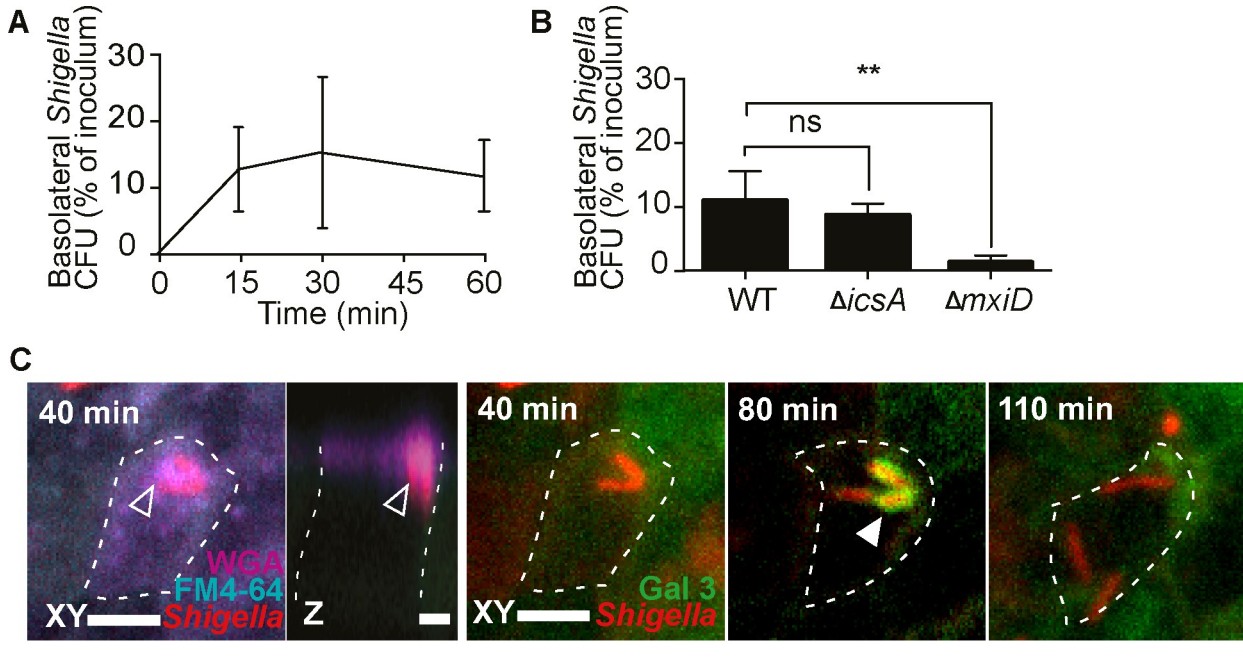

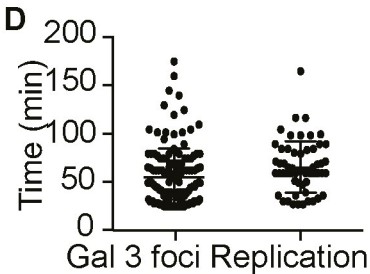

**Fig 2. *S. flexneri* follows a bimodal itinerary within M cells.** (A) *S. flexneri* translocates rapidly to basolateral compartments (t15, 30 *n* = 5, t60 *n* = 6). (B) *S. flexneri* transcytosis is dependent on bacterial invasion and independent of ABM at 1h p.i.,(WT *n* = 5, Δ*icsA* and Δ*mxiD* *n* = 4, *P <0.01). (C) *S. flexneri* induces apical membrane ruffling at 40 min p.i. in a galectin 3-eGFP reporter M cell (delineated by a dashed line), visible in the FM4-64 and WGA channels (empty arrows), before rupturing its vacuole at 80 min p.i. visible by the formation of a galectin 3 cage around the bacteria, and cytosolic replication at 110 min pi, visualized by time-lapse microscopy. Representative time-lapse (n = 3), at different Z-planes (based upon the best view of the different infection events), in the same XY field. Scale bar, 5 μm. See also video S1. (D) Quantification of the onset of *S. flexneri* vacuolar rupture and replication within M cells imaged live (Gal 3 foci *n* = 139, replication *n* = 51, data from three independent experiments).

comet tails were observed within WGA-positive M cells. These motile bacteria also formed membrane protrusions (visible through FM4-64) while spreading into neighboring cells starting from 130 ± 38 min p.i. (Fig 3C and 3D, S2 Movie). These observations provided direct evidence that *S. flexneri* ABM within M cells drives its spread to neighboring enterocytes.

## *L. monocytogenes* subverts M cell transcytosis and spreads directly from the M cell cytosol to neighboring enterocytes via ActA

*L. monocytogenes* shares with *S. flexneri* a tropism for M cells and a similar lifestyle within target cells including ActA-mediated ABM [14,29]. These resemblances led us to compare their behavior with co-culture M cells. In accordance with previous *in vitro* studies, *L. monocytogenes* did not translocate through the co-cultures at 1 hour p.i. (Fig 4A and 4B), suggesting

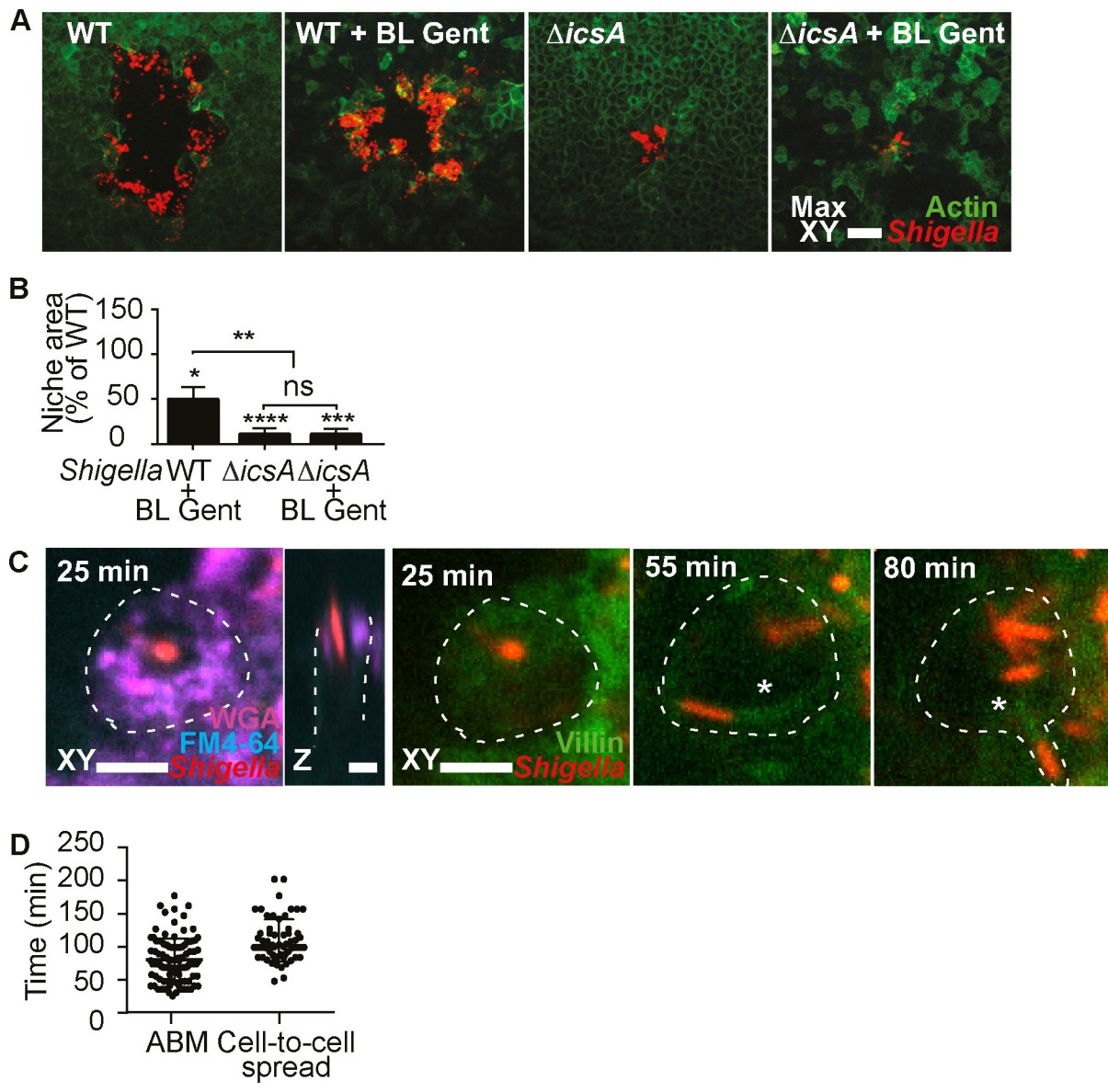

**Fig 3. *S. flexneri* spreads directly from the M cell cytosol to neighboring enterocytes via IcsA.** (A) *S. flexneri* infection plaques formed in the indicated conditions and strains, visualized by confocal microscopy at 21h p.i.. WT = wild-type, gent = basolateral gentamicin. Scale bar, 50 μm. (B) Quantification of infection plaque areas for the indicated conditions, normalized to WT. The statistical significance of differences between conditions and WT were assessed by one sample t tests, and differences between conditions were assessed by one-way ANOVA and multiple comparisons (WT $n$ = 61, WT + BL Gent $n$ = 40, $\Delta icsA$ $n$ = 141, $\Delta icsA$ + BL Gent $n$ = 121, WT and $\Delta icsA$ data from five independent experiments, + BL Gent data from three independent experiments, (*$p < 0.05$, **$p < 0.01$, ****$p < 0.0001$) (C) *S. flexneri* induces apical membrane ruffling at 25 min p.i. in a villin-eYFP reporter M cell (delineated by a dashed line), before developing ABM at 55 min p.i., visible by the villin-positive comet tail (asterisk), leading to the development of a membrane protrusion visible by FM4-64 labeling and direct spread at 80 min p.i. into a neighboring cell. Scale bar, 5 μm. See also video S2. (D) Quantification of the onset of *S. flexneri* ABM and cell-to-cell spread from M cells (ABM $n$ = 93, Cell-to-cell spread $n$ = 46, data from three independent experiments).

another route for FAE translocation *in vivo* [30,31]. In contrast, we found that non-motile *ΔactA* mutants translocated in conditions of preserved barrier integrity, indicating that *L. monocytogenes* evades M cell transcytosis through an ActA-dependent mechanism (Fig 4A and 4B, S6B Fig). Interestingly, we noted that classical virulence effectors for vacuolar rupture,

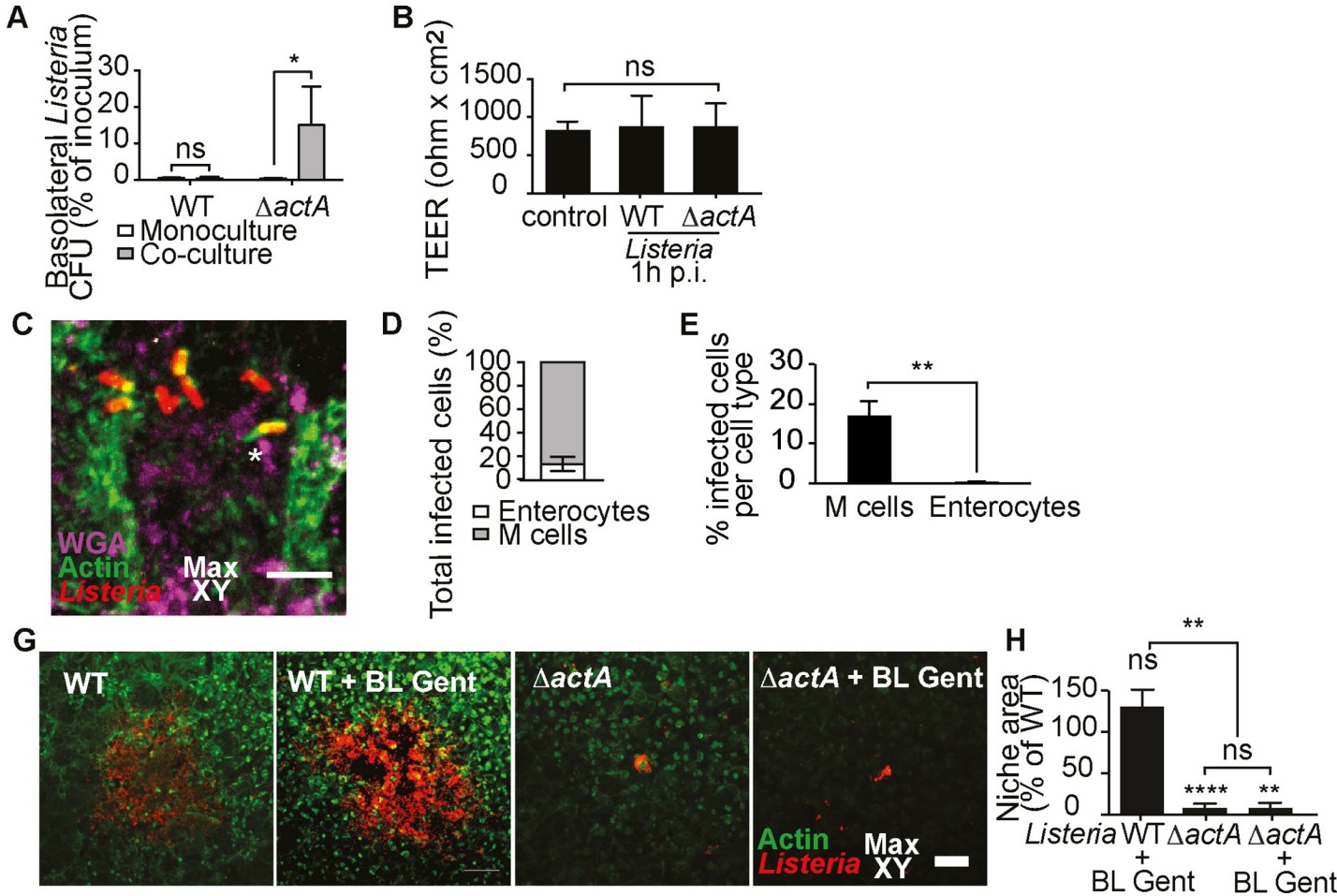

**Fig 4. *L. monocytogenes* subverts M cell transcytosis and spreads directly from the M cell cytosol to neighboring enterocytes via ActA.** (A) *L. monocytogenes* evades translocation through co-cultures via ActA at 1h p.i., (monocultures WT and Δ*actA* n = 3, co-cultures WT n = 9, Δ*actA* n = 7, *P < 0.05) (B) Co-culture barrier integrity is preserved with the indicated strains at 1h p.i.. (non infected control n = 6, challenged n = 4) (C) *L. monocytogenes* develops actin comet tails within M cells at 3h p.i.. Representative image (n = 211, data from three independent experiments) Scale bar, 50 μm. (D-E) *L. monocytogenes* preferentially targets M cells versus enterocytes at 3h p.i. (n = 235, data from three independent experiments, *P <0.01). (G) Representative images of *L. monocytogenes* infection plaques formed under the indicated conditions and strains in the co-cultures, visualized by confocal microscopy at 21h p.i.. (H) Quantification of infection plaque areas under the indicated conditions, normalized to WT. The statistical significance of differences between conditions and WT were assessed by one sample t tests, and differences between conditions were assessed by one-way ANOVA and multiple comparisons (WT n = 39, WT+ BL Gent n = 34, Δ*actA* n = 39, Δ*actA* + BL Gent n = 35, data from four independent experiments, (*p < 0.05, **p < 0.01, ****p < 0.0001).

LLO, pIcA and pIcB, had no impact on M cell transcytosis subversion, suggesting an alternative mechanism for vacuolar rupture in this cell-type (S7 Fig).

We evaluated the interaction of *L. monocytogenes* with M cells by confocal imaging at 3 hours p.i.. At this time point, we identified 80% of the infected cells as M cells based on their apical morphology and WGA staining pattern, thus establishing this cell type as initial target for *L. monocytogenes* in our model. Furthermore, we frequently observed events of bacterial actin comet tail formation (Fig 4D–4F), thus demonstrating the promotion of cytosolic ABM by *L. monocytogenes* within M cells, as observed with *S. flexneri*.

We assessed the process of *L. monocytogenes* dissemination within the FAE model. Large plaques of infection were formed by the pathogen at late time-points of infection. Presumably due to the capacity of *L. monocytogenes* to evade M cell transcytosis, basolateral bacteria were not involved in bacterial propagation; whereas ActA-dependent ABM was required for

epithelial dissemination of the pathogen, similar to *S. flexneri* (Fig 4G and 4H, S6C Fig, S1 Appendix).

In summary, we uncovered the shared ability of *L. monocytogenes* to subvert M cell transcytosis via ActA and promote ActA-dependent actin-based motility for dissemination within the FAE.

## Discussion

M cells play a central role in the onset of many enteric infections and the induction of mucosal immunity in the GALT, but advanced host-pathogen interactions studies with this cell-type have been difficult to perform due to their rarity *in vivo* and challenging culture *in vitro*. Here, we report a novel workflow allowing the detailed characterization of invasive enteric pathogen interactions with M cells. Focusing on two major gastrointestinal pathogens, *S. flexneri* and *L. monocytogenes*, we extended an established human *in vitro* FAE model to physiological infections, expression of fluorescent reporters and visualization by long-term time-lapse microscopy. This approach allowed us to reveal that *S. flexneri* and *L. monocytogenes* can subvert the antigen-sampling function of M cells by promoting vacuolar rupture and cytosolic invasion. In the M cell cytosol, invading pathogens are able to promote direct actin-based M cell-to-enterocyte spread, a novel route promoting epithelial dissemination of the infection from this entry point. Our results suggest that intracellular pathogens may have evolved their intra- and intercellular lifestyle to avoid exposure to basolateral immune hubs from M cells *in vivo*, a process that may be important for their ability to evade the induction of adaptive immunity.

Although alternative routes for intestinal invasion have been reported during infections by intracellular pathogens [7,32,33], their common early association with the FAE overlying lymphoid follicles has consistently been observed following oral contamination of human patients, macaques and mice [24,34,35]. At the FAE, the predominant role of M cells for initial pathogen uptake has further been underscored *in vitro* and *in vivo*, based on ligated loop models [4,5,9,14,36]. Until now, M cells have mostly been studied for their role in particle translocation, but their outstanding biology during pathogen invasion has remained obscure. In particular, *S. flexneri* and *L. monocytogenes* interaction studies with M cells have mainly been restricted to a handful of *in vivo* analyses, essentially validating their initial targeting of this cell-type, but providing poorly quantifiable data on their behavior within this host [5,14,37]. To clarify the nature of *S. flexneri* and *L. monocytogenes* interactions with M cells, we performed physiological infections of an established human *in vitro* FAE model that has been broadly used in the scientific community, and which we characterized by single-cell transcriptomics [21–23]. This approach allowed us to obtain the complete picture of *S. flexneri* and *L. monocytogenes* infections, from their apical onset in M cells, to basolateral translocation and dissemination at the FAE. Moreover, we introduce a new protocol for bacterial and host expression of fluorescent reporters and M cell host-pathogen interaction visualization by long-term confocal time-lapse microscopy. Our workflow provides a new step forward in the resolution of transient infection events and their dynamics in M cells such as vacuolar rupture and new dissemination pathways. The ability to track host molecules and invading bacteria in M cells over the course of infection is pivotal to decipher new mechanisms of infection at play, which cannot be captured by end-point assays in this rare cell-type.

The unique architecture of the M cell trafficking compartment, designed for antigen sampling, and its impact on host-pathogen interactions is unclear. While it has been established that M cells can efficiently and rapidly translocate various types of antigens to basolateral immune inductive hubs, it is not well known whether their apparent flexibility to cytoskeletal rearrangements and vesicular trafficking confers susceptibility or resistance to subversion by

intracellular pathogens [38]. Thus far, M cells have been perceived as gateways at the FAE for *S. flexneri* and *L. monocytogenes* into the intestinal mucosa, offering them a direct route from the intestinal lumen to deeper host tissues. A paradigm has emerged, that similarly to other microbes and particles, these pathogens remain encased within a vacuole throughout their transcytotic journey across M cells [5,6,15]. In this work we report that *S. flexneri* develops a bimodal lifestyle within M cells leading to rapid transcytosis or slower escape to the cytosol by vacuolar rupture; while *L. monocytogenes* evades transcytosis completely via the virulence factor ActA, developing its niche within the M cell cytosol. With *S. flexneri* we show that transcytosis occurs within 15 minutes, and is dependent on the functional expression of the T3SS, known to induce massive recruitments of host endomembrane involved in vacuolar rupture of invading bacteria [39,40]. In line with this phenomenon, we noted the presence of apical membrane ruffles around slower invading bacteria in M cells by time-lapse microscopy (Fig 2C and 2D, S1 Movie). We raise the possibility that the T3SS-dependent diversion of the M cell endomembrane compartment is impaired or delayed in this cell-type due to its extensive solicitation for antigen sampling [41]. This needs further investigations, as it could explain the inability of early interacting pathogens to rupture their vacuole and the delay in vacuolar rupture which is observed with cytosolic bacteria in M cells (74 ± 33 min) compared to enterocytes (around 20 min in our hands and within 15 to 30 min as reported in the literature) [11,27].

In contrast, we show that *L. monocytogenes* transcytosis subversion is dependent on the virulence effector ActA, which has been shown to be involved in a specific apical internalization pathway into epithelial cells [42]. We were unable to perform time-lapse microscopy on *L. monocytogenes*, as it appeared to be very sensitive to blue light in our hands [43], however we surmise that this route may be critical for the downstream ability of *L. monocytogenes* to rupture its vacuole within M cells. Along these lines, we noted that the classical vacuolar rupture effectors LLO, pIcA and pIcB do not play a role in transcytosis evasion by the pathogen, suggesting a different vacuolar rupture mechanism in M cells. The observed ability of *L. monocytogenes* to subvert M cell transcytosis at the FAE suggests that the pathogen may target dendritic cells directly for translocation to deeper tissues *in vivo*, as previously proposed [31].

Finally, we resolved the shared ability of *S. flexneri* and *L. monocytogenes* to respectively initiate IcsA and ActA-dependent ABM within the M cell cytosol for epithelial dissemination (Figs 3E and 4G). Of note, it would not have been possible to capture this route *in vivo*, as this hasn't been achieved with other cell-types until now. By time-lapse microscopy, we observed the appearance of actin comet tails at the pole of *S. flexneri* within 90 ± 46 min p.i., a larger time frame than that observed for vacuolar rupture, as expected and reported in enterocytes [27]. This was followed by the initiation of cell-to-cell spread at 130 ± 38 min p.i..

Moreover, we revealed that this novel direct M cell-to-enterocyte route was essential for the propagation of the epithelial infection in our system; in contrast with the previous assumption that *S. flexneri* reinvade the epithelium from the basolateral side after M cell transcytosis [6]. In terms of dynamics, this direct route may fit better with the timeline of epithelial infection progress *in vivo*, as it avoids potentially lengthy encounters with underlying immune cells and the subsequent repetition of the invasion cycle in enterocytes [9].

It has remained mysterious why *S. flexneri* and *L. monocytogenes* induce poorly protective immune responses *in vivo*, despite their early interactions with M cells at inductive immune hubs of the intestinal mucosa [13,44]. We propose that their ability to subvert M cell transcytosis and promote lateral IcsA- and ActA-driven M cell-to-enterocyte spread provides a potential mechanism for their evasion of adaptive immunity. Indeed, this route may allow them to evade exposure to basolateral lymphoid follicles *in vivo* and impede the efficient induction of immune responses. This scenario would be in line with published results from vaccine trials

showing the elaboration of fully protective immune responses with non-motile Δ*icsA* and Δ*actA* mutants [44,35].

Taken together, our findings could represent a paradigm shift in the early mechanisms of mucosal penetration by intracellular pathogens, as they suggest that *S. flexneri* and *L. monocytogenes* may subvert M cell transcytosis to underlying tissues and remain within the epithelium during early *in vivo* dissemination. These results would suggest that bacteria could have evolved hallmarks of their intracellular lifestyle under the pressure of becoming exposed to the GALT inductive system within M cells. We propose that M cell transcytosis subversion mechanisms may be pivotal during evasion of adaptive immunity by intracellular pathogens, and hope that their better comprehension can foster ideas for new mucosal therapies. To conclude, this study highlights the importance of advanced M cell host-pathogen investigations to understand early steps of mucosal colonization and immune evasion.

## Materials and methods

### Cell culture

Caco-2 cells RRID:CVCL_0233 and CVCL_1096ref were grown in DMEM supplemented with 10% decomplemented FCS, 1% HEPES, 1% non-essential amino acids and 1% penicillin/streptomycin (100 units/ml and 100 μg/ml, respectively) at 37˚C and 10% CO2. Raji B cells RRID:CVCL_0511were grown in RPMI supplemented with 10% decomplemented FCS, 1% non-essential amino acids and 1% penicillin/streptomycin (100 units/ml and 100 μg/ml, respectively) Caco-2 and Raji B cells are of human male origin. The Caco-2 cells were authenticated by their ability to differentiate spontaneously into a polarized epithelium and to transdifferentiate into M cells. The Raji B cells were authenticated by their ability to induce the transdifferentiation of Caco-2 cells into M cells.

### Bacteria

The following M90T 5a *S. flexneri* strains were used throughout the assays except for time-lapse microscopy: the wild-type strain, the mutant strains for effectors MxiD (*ΔmxiD*) [25], IcsA (*ΔicsA*) [12] transformed with pGG2(Ampr)-dsRed [45]. For *S. flexneri* IcsA complementation experiments, the previously published SC560 ΔicsA and SC561 ΔicsA + complemented strains were used [46].

For time-lapse microscopy, the wild-type strain M90T dsRed AfaI expressing the adhesin AfaI was used. Briefly, a pBR322-(Ampr)-AfaI-dsRed plasmid was used to transform the wild-type M90T 5a *S. flexneri* strain, constructed by ligating the dsRed coding sequence from pGG2 (Ampr)-dsRed into pBR322(Ampr)-AfaI [47].

The following *L. monocytogenes* EGD strains (kindly provided by P. Cossart) were used: the wild-type (BUG600), the mutant strain for effector ActA (Δ*actA*, BUG 2140) and the mutant strain for LLO (Δ*actA*, BUG 2132). For *L. monocytogenes* ActA complementation, the BUG 2140 strain was transformed with the plasmid p*actA3* containing the *actA* structural gene and immediate upstream promoter, thus generating the complemented BUG 2140 EGD Δ*actA* + strain [29]. *L. monocytogenes* EGD-e PrfA* wild-type (BUG3057) and the triple mutant strain for LLO, pIcA and pIcB (Δ*hly* Δ*pIcA* Δ*pIcB*, BUG3648) were kindly provided by Javier Pizarro-Cerdà.

For *S. flexneri* infection experiments, bacteria were grown overnight (ON) in tryptic casein soy broth (TCSB), supplemented with ampicillin (50 μg/ml) or spectinomycin (50 μg/ml) depending on the resistance of the strains, at 37˚C with shaking at 220 rpm and subsequently subcultured at a 1/100 dilution for 2.25 h to a measured optical density at 600 nm (OD600)

above 0.3. For *L. monocytogenes* infection experiments, bacteria were grown ON in liquid bovine heart infusion medium (BHI) at 37°C with shaking at 220 rpm.

## Culture of the intestinal human FAE model

To generate polarized monocultures of Caco-2 cells or co-cultures of Caco-2 and M cells, $10^6$ Caco-2 cells were seeded in the top compartment of transwell inserts, on 24mm polycarbonate membranes with 3 μm pores (Corning). The cultures were differentiated for 14–15 days with medium replacement every 2–3 days. To generate co-cultures of Caco-2 and M cells, $0.5 \times 10^6$ human Raji B cells (kindly provided by A. Alcover) were added to the bottom compartment of the transwell on day 11 and kept until the end of the differentiation process, inducing the transformation of a subset of Caco-2 cells into M cells. Transepithelial resistance (TEER) of the monolayers was controlled to assess the differentiation of the monolayers.

## Generation of stable Caco-2 reporter cell lines

Two stable Caco-2 C2BBe1 (ATCC CRL-2102) cell lines expressing galectin-3-eGFP and villin-eYFP respectively, were generated using the pLenti6/V5 Directional TOPO Cloning Kit (Invitrogen) with the coding sequences from the pEGFP-galectin-3 plasmid [26][48](primers: 5'- CGCGCGCCGCGGTTATATCATGGTATA- 3' and 5'–GAGTAAACTAGTATGGTGAG CAAGGGC– 3') and the pEYFP-villin plasmid [28] (primers: 5'- CATGTCACTAGTATGGT GAGCAAGGGCGAG- 3' and 5'-GAGTAACCGCGGTCAAAATAGTCCTTTTC- 3'). The resulting fragments were cloned into pLenti6/V5-D-TOPO(r) (K495510, Invitrogen) using SacII and SpeI and lentiviruses were produced in HEK cells according to the manufacturer protocol. Caco-2 cells were transduced and single clones were selected and expanded after isolation using a BD FACS sorter. The differentiation and M cell transformation capacity of the clones, in conditions of preserved expression of the genetically encoded fluorescent reporters, was controlled by TEER measurements, confocal analysis of apical WGA binding and fluorescent reporter expression patterns (typically, a diffuse cytosolic signal for galectin-3-eGFP and a cortical signal for villin-eYFP), respectively.

## Infection assays

Mono- and co-cultures on transwells were washed in EM medium and equilibrated 1h at 37°C and cooled to room temperature (RT). Bacteria were washed in phosphate-buffered saline (PBS) once for *S. flexneri* and four times for *L. monocytogenes*. The bacteria were then resuspended in EM at a multiplicity of infection (MOI) of 10 to 30 bacteria per cell for *S. flexneri* and 5 to 15 bacteria per cell for *L. monocytogenes*. To synchronize infection, the bacteria were allowed to settle in the upper transwell compartment on the apical face of the monolayers for 15 min at RT, before initiating the infection by switching the temperature to 37°C. At 1h postinfection (p.i.), the apical face of the monolayers was extensively washed to remove extracellular bacteria and treated for 1h at 37°C with apical gentamicin (100 μg/mL) to kill residual extracellular bacteria and prevent apical infections *de novo*. The apical solution was then changed to a solution of gentamicin (10 μg/mL) and decomplemented FCS (10%) for further incubation at 37°C and 10% CO2. For basolateral gentamicin treatments, the lower compartment of the transwells was supplemented with gentamicin (30 μg/mL) to treat the basolateral face of the monolayers throughout the whole infection time course.

## Translocation assays

For bead translocation assays, mono- and co-culture monolayers on transwells were washed in EM medium (120 mM NaCl, 7 mM KCl, 1.8 mM CaCl2, 0.8 mM MgCl2, 5 mM glucose, and 25 mM HEPES at pH 7.3) and equilibrated 1h at 37˚C and then cooled to 4˚C. A suspension of 0.02 μm yellow-green carboxylated polystyrene beads (F8787, Molecular Probes) in EM medium ($1.5 \times 10^{12}$ beads/transwell) was then added in the upper compartment of the transwells to the apical surface of monolayers for 60 min at 4˚C and switched at 37˚C for a further 60 min of incubation. Aliquots from the basolateral compartment of the transwells were taken at different time-points and fluorescence intensities were measured using an Infinite 500 TECAN fluorimeter at excitation and emission wavelengths of 485 ± 20 nm and 520 ± 20 nm, respectively. Four independent experiments were performed and all samples were run in duplicates and measured against a blank and a standard curve.

For bacterial translocation assays, a minimum of three independent experiments were performed per condition and strain. Bacteria from the apical inoculum and translocated bacteria in the bottom compartment of the transwells were collected at different time-points and serially diluted. Colony forming unit (CFU) counts were performed by plating the different dilutions of bacteria on TCSB and BHI agar plates for *S. flexneri* and *L. monocytogenes*, respectively.

To assess the impact of superglue treatment on the translocation ability of the co-cultures during live imaging, four drops of superglue were added on the surface of co-culture monolayers before addition of the bacterial mix and execution of the bacterial translocation assay.

## Indirect Immunofluorescence

For immunofluorescence labeling of infected monolayers, infections on transwells were stopped by placing the monolayers at 4˚C, excising the transwell membranes, washing the excised monolayers on membranes in PBS and plunging them in 4% paraformaldehyde (PFA) at RT for 10 min. Fixed samples were blocked in 5% BSA for 30 min at RT and permeabilized in 0.5% Triton for 5 min at RT. Monolayers were incubated with polyclonal antibodies against *S. flexneri* (generated and kindly provided by A. Phalipon) or L. monocytogenes (antibody R11) at 1:500 and 1:200 in blocking buffer for 1h 30 at RT, respectively. After washing the monolayers in blocking buffer, secondary anti-rabbit Cy3 (A10520, Thermofisher) antibodies at 1:200 were incubated together with phalloïdin Alexa Fluor 488 (A12378, Thermofisher) at 1:100 to label actin and DAPI (D1306, Thermofisher) at 1:100 in blocking buffer to label nuclei for 1h at RT. The monolayers were then mounted in Prolong Gold Antifade reagent (P36931, Thermofisher).

Histochemical characterization of the FAE model.

Wheat-germ agglutinin (WGA), which binds with high affinity to the sialic-acid and N-acetylglucosamine residues present in M cells, and the M cell specific FimH receptor glycoprotein 2 (GP2) were used for identifying M cells. Conversely, Ulex Europaeus Agglutinin 1 (UEA-1), which binds to α-L-fucose and sucrase isomaltase (SI) were used to identify differentiated enterocytes (S1 Appendix). For WGA, SI and UEA-1 immunofluorescence labeling of control monolayers on transwells, the monolayers were incubated live with WGA Alexa Fluor 633 (W21404, Thermofisher) at 1:200 in EM for 5 min at 37˚C and washed in PBS at RT before excision of the transwell membranes and fixation of the excised monolayers on membranes in absolute methanol for 5 min at -20˚C. The monolayers were then washed and incubated with antibodies against sucrase-isomaltase (HPA011897, Sigma) at 1:500 in PBS for 1h 30 at RT. Afterwards, monolayers were washed and permeabilized 5 min in 0.5% Triton and incubated with the secondary antibody anti-rabbit Cy3 at 1:200, the lectin UEA1 FITC (L9006, Sigma) at

1:500 and DAPI at 1:100 in PBS to label nuclei, for 1h at RT. The monolayers were then mounted in Prolong Gold Antifade reagent.

For GP2 immunofluorescence labeling of control monolayers on transwells, monolayers were washed in PBS at RT before excision of the transwell membranes and fixation of the excised monolayers in 4% paraformaldehyde (PFA) at RT for 10 min. Monolayers were then washed and blocked in 5% BSA for 30 min at RT. Monolayers were incubated with antibodies against GP2 (D277-3, MBL) at 1:100 in blocking buffer for 1h 30 at RT. The monolayers were then washed in blocking buffer and incubated with the secondary antibody anti-mouse Rhodamine (31660, Thermofisher) for 1h. Afterwards, monolayers were washed and permeabilized in 0.5% Triton for 5 minutes before incubation with phalloïdin Alexa Fluor 488 and DAPI in PBS for 1h at RT. Finally, monolayers were mounted in Prolong Gold Antifade reagent.

To assess the impact of our live imaging workflow on the histochemical properties of the co-culture monolayers, co-cultures were excised live and glued upside-down on the surface of a well (as detailed below) and incubated at 37˚C. After 1h the samples and control non-treated monocultures were fixed and characterized by immunofluorescence labeling of GP2 and WGA binding.

## Time-lapse microscopy

For time-lapse imaging of *S. flexneri* infection, monolayers on transwells were incubated *in situ* with the M cell marker WGA Alexa Fluor 633 at 1:200 in EM for 5 min at 37˚C, and washed in EM at RT. Bacteria were incubated in EM supplemented with the eukaryotic membrane dye FM4-64 (2 μM, T3166, Thermofisher) and allowed to settle in the upper compartment of the transwells on the apical face of the monolayers for 15 min at RT, followed by incubation for 15 min at 37˚C.

The transwell membranes were then excised live and the obtained monolayers on the membranes were glued upside down on the surface of a ibiTreat μ-slide well (Ibidi) with 0.5 μL of liquid super glue (Loctite, 577093), incubated in EM supplemented with FM4-64 and imaged at 37˚C (S4 Fig). As *S. flexneri* is non-motile, poorly adherent and would detach by gravity from the apical face of the monolayers in this configuration, M90T dsRed/AfaI bacteria were used, at an MOI lowered to 5 to 15 bacteria per cell. Moreover, we observed that the M cell marker WGA is transcytosed by co-cultured M cells while performing the experiments, therefore constituting a transient marker (as previously observed *in vivo* [49]). In consequence, the WGA channel was only acquired at early time points to allow the identification of M cells. During the course of image acquisitions, at 1h p.i., the incubation medium was changed to a solution of gentamicin (100 μg/mL) and FM4-64 to kill extracellular bacteria (apical and basolateral). This solution was then changed at 2h p.i., to a solution of gentamicin (10 μg/mL), decomplemented FCS (10%) and FM4-64 during acquisition and maintained for further acquisition at 37˚C.

Time-lapse microscopy was performed using an UltraView spinning disc confocal microscope (Perkin Elmer) with a 40x [1.3 NA] oil objective. Every 3 to 10 minutes, a z-stack of 70 planes with 0.6 μm z steps was acquired sequentially in 3 channels for 7h, with a final timepoint acquired at 21h, using a 488 nm and a 561 nm laser line. Fluorescence emission was detected with 525 (W50) nm, 615 (W70) nm and 705 (W90) nm filters respectively. At early time-points, an additional channel was used for the acquisition of WGA binding of M cells using a 633 nm laser and detecting fluorescence emission with a 705 (W90) nm filter. Infection was stopped by placing the monolayers at 4˚C, excising the transwell membranes and plunging the monolayers in 4% paraformaldehyde (PFA) at RT for 10 min.

## Image analysis

To quantify M cell populations within the co-cultures, three independent experiments were performed and 30 fields of view of control M cell containing co-cultures were randomly chosen and analyzed. M cells were identified by the apical loss of a dense and organized brush border, visualized by phalloïdin staining of actin, manually counted and normalized to the total number of cells. For histochemical identification of M cells within the co-cultures, four independent experiments were performed and 20 fields of view of control M cell containing co-cultures were randomly chosen and analyzed. M cells were identified by the apical expression of GP2 and WGA by immunofluorescence and fluorescence labeling respectively, manually counted and normalized to the total number of cells.

To compare the apical surface characteristics of Caco-2 monocultures versus M cell containing co-cultures, at least four independent experiments were performed for each marker and at least 20 fields of view of control Caco-2 monocultures and M cell containing co-cultures were randomly chosen and analyzed. The number of cells positive for the apical expression of each marker was manually counted. The fold change number of positive cells for each marker was assessed co-cultures versus monocultures.

To assess the impact of excision, inversion and supergluing of the co-cultures in imaging wells, 3 independent experiments were performed and 30 fields of view of control Caco-2 monocultures and treated M cell containing co-cultures were randomly chosen and analyzed. The number of cells positive for the apical expression of each GP2 and WGA was manually counted. The fold change number of positive cells for these markers was assessed in the treated co-cultures versus control monocultures.

To evaluate the areas of infection niche formation, for each comparison against WT, a minimum of 3 experiments were performed per strain and a minimum of 30 niches were randomly chosen and analyzed. Maximum 3D projections of entire Z-stacks were assembled, ROIs were manually delineated around peripheral infected cells identified by actin and intracellular bacteria, and areas were measured in Fiji or Icy. Mean areas for each strain were normalized to the mean area of WT in each experiment.

To evaluate the cell-type preferentially targeted by *L. monocytogenes* during early co-culture infection, three experiments were performed and a minimum of 50 infected cells were randomly chosen and analyzed per experiment. Infected M cells were identified by the paucity of apical microvilli expression, visualized by phalloïdin staining of actin, as well as the apical binding of WGA. The relative proportion of infected M cells and enterocytes was deduced after visual identification and manual counting. The ability of *L. monocytogenes* to perform actin-based motility within host cells was assessed by the visualization of comet tails, identified by phalloïdin staining of actin, consisting in assymetrically distributed F-actin extending from one pole of the bacterium.

## Live-Cell image analysis

Image series were visually analyzed. All events were monitored using the timing post-infection, i.e. the time at which co-cultures challenged with apical bacteria were placed at 37˚C. The perimeters of the individual cells were manually delineated after visual assessment of each fluorescent channel in the XY, in the Z and over time (the FM-464 membrane dye provides a peripheral marker by staining the cellular plasma membrane, galectin-3-eGFP provides a cytosolic marker which visually complements the delineation of the periphery of the cell when vacuolar rupture isn't occurring, villin-eYFP is enriched at the cortex of the cell, thus providing another visual marker of the cellular periphery). M cells were identified by the apical binding of WGA. Vacuolar rupture was determined by the appearance of a galectin-3 signal around

the bacteria. Initiation of intracellular replication was determined by the onset of binary fission in the bacteria. Intracellular motility was determined by the recruitment of villin at one pole of the bacteria. Cell-to-cell spread initiation was determined by the crossing of the basolateral membrane between the host and a neighboring cell, leading to its penetration by the bacteria. For the dynamic characterization of these events, three independent experiments were performed and a minimum of 30 events was analyzed per condition.

### Single cell transcriptomics

To extract the gene expression profiles of the M cells in our model, single cell transcriptomes of the human co-culture Caco-2 enterocytes and M cells were compared to control monocultures of Caco-2 enterocytes and control Raji B cells. One day before the experiment, control monocultures of Caco-2 enterocytes and co-cultures of Caco-2 enterocytes and M cells were stained respectively with a 5 µM solution of CellTrace CFSE and a 1 µM solution of CellTrace Far Red (C34554 and C34564, Thermofisher) in PBS, following the manufacturer's instructions. The following day, prelabeled monocultures of Caco-2 enterocytes and co-culture Caco-2 enterocytes and M cells were dissociated into suspensions of single cells by Trypsin-EDTA treatment (0.25%, Thermofisher) for 10 min at 37˚C, followed by vigorous resuspension and washing in a solution of ice-cold EM and decomplemented FCS (10%). In parallel, a fresh culture of proliferating Raji B cells was harvested and resuspended in a solution of ice-cold EM and decomplemented FCS (10%). The three suspensions of cells were then mixed in suspension buffer (101–0063, Fluidigm) in the following proportions: 2.5% of Raji B cells, 5% of monoculture Caco-2 enterocytes and 92.5% of co-culture Caco-2 enterocytes and M cells to the final concentration of 0.4 x$10^6$ cells/mL.

A high concentration of co-culture cells was used in order to collect as many M cell transcriptomes possible (as M cells represent only 5–10% of the co-culture cells) and a lower concentration of Raji B cells was used for technical reasons detailed below. The cellular mix was loaded onto a medium HT C1 IFC chip (101–0221, Fluidigm) allowing the random capture of up to 800 cells of 10–17 µm diameter into individual capture sites compatible with microscopy visualization and preparation of single cDNA libraries for RNA sequencing. As Raji B cells are smaller in size than the epithelial cell types, with a higher probability to be captured by the chip inlets, a lower proportion of Raji B cells was loaded.

The successful capture of single cells was controlled by epifluorescence microscopy using a Zeiss Definite Focus inverted microscope. Each capture site was acquired with a 10x/0.45 Plan Apochromatic air objective using a bright field channel, a Colibri LED excitation system at 470 nm and a fluorescence light bulb combined with an excitation filter at 605–645 nm. Fluorescence emission was detected using a pass-brand detection filter at 500–550 nm and an Omega filter XF110-2 at 668–723 nm. This allowed the visual assessment of the size and fluorescence of single captured cells for identification of control Caco-2 enterocytes, control Raji B cells and co-culture cells.

Following this, captured cells were lysed, cellular mRNA and RNA spikes (ERCC, Thermofisher) were reverse-transcribed and cDNA with specific barcodes for each cell was preamplified on the C1 system. The cDNA libraries were prepared for sequencing using a Nextera kit (fc-131-1096, fc 131–1001, Illumina), which provided a supplementary index for each single cell cDNA, and assessed for quality and concentration using BioAnalyzer (Agilent) and Qubit (Thermofisher) technologies. Pools of 20 individual cDNA libraries were normalized to 2 nM, denatured and diluted to 9 pM for loading on the flowcell. The sequencing was performed by pair-end on a HiSeq 2500 sequencer (Illumina). Genetic material from capture sites with more than one cell (controlled by microscopy), or of poor quality (controlled by BioAnalyzer and

Qubit) as well as single cells with an average and median number of reads below 72285 and 44504, respectively, were discarded- narrowing down the number of analyzed cells to 63 (8 control Raji B cells, 5 control Caco-2 enterocytes and 50 co-culture Caco-2 enterocytes and M cells). Raw sequencing reads were analyzed using the RNA-seq pipeline from Sequana [50]: mapping was performed on the human hg19 reference genome from UCSC and counting was done using the gencode annotation (v28). Resulting count data were then analyzed using R version 3.5.1 [51] and the Bioconductor package DESeq2 version 1.20.0 [52] with default parameters. For principal component analysis of the data, count data were made homoscedastic using the DESeq2 Variance-Stabilizing Transformation before performing principle component analyses (PCA). Inertia percentages associated with the two first principal components were provided in the axes' labels.

## Quantification and statistical analysis

Statistical analysis was performed with the software GraphPad Prism v6.0. For each data set, a Shapiro-Wilk test was used to evaluate if the values were normally distributed. The statistical significance of differences between experimental conditions with normal Gaussian distribution were analyzed by two-tailed unpaired t tests or one-way ANOVA with Sidak/Dunnett multiple comparisons. For normalized data (percentages or fold-change versus control), a one-sample t test was performed against the control value (100 or 1, respectively). If at least one condition presented a non normal distribution, nonparametric tests were used: Mann Whitney tests or Kruskal-Wallis for multiple comparisons were performed. Unless otherwise stated, all experiments were performed at least three times and the data were presented as mean ± SD. The statistical details of experiments can be found in the figures, figure legends and method details sections. The definition of statistical significance was $P < 0.05$, $^{*}p < 0.05$, $^{**}p < 0.01$, $^{***}p < 0.001$, $^{****}p < 0.0001$.

## Supporting information

**S1 Fig. Characterization of the apical morphology and surface composition of M cell containing co-cultures. (A)** Representative image of co-culture M cells identified by the apical expression of GP2 and the disorganized morphology of the apical brush border visible by actin staining (data from at least three independent experiments) Scale bar, 10 μm. **(B)** Representative images of mono- and co-culture apical binding of M cell specific WGA and enterocyte-specific UEA1; and apical expression of enterocyte-specific SI (data from at least four independent experiments) Scale bar, 10 μm. **(C)** Quantification of the fold-change difference in cells positive for M cell- and enterocyte expression markers (GP2/WGA and SI/ UEA-1, respectively) in co-cultures versus monocultures (GP2 n = 15, WGA n = 35, SI n = 25, UEA1 n = 20) Data is from at least four independent experiments. Data are mean ± s.d. ($^{****}p < 0.0001$). (TIF)

**S2 Fig. Characterization of the transcytotic ability of M cell containing co-cultures. (A)** Time-course of 20 nm carboxylated polystyrene bead translocation rates across mono- and co-cultures, with a temperature switch from 4˚C to 37˚C. Data is from four independent experiments. **(B)** Transepithelial electrical resistance (TEER) of control mono- and co-culture monolayers. Data are from five independent experiments. Data are mean ± s.d. ($^{****}p < 0.0001$). (TIF)

**S3 Fig. Single cell transcriptomic analysis of co-culture M cells. (A)** Scheme of the workflow for isolation, identification and RNA sequencing of single co-culture M cells. Single cell suspensions of Raji B cells, CFSE-labeled monoculture Caco-2 cells and Far Red-labeled co-

cultured Caco-2 and M cells are prepared and loaded as a mix onto a HT C1 chip (Fluidigm). The C1 system randomly captures single cells into individual capture sites. Each capture site is acquired at the fluorescence microscope to validate and identify single captured cells. The C1 chip is run for lysis, mRNA reverse transcription (RT) and pre-amplification of the cellular cDNA. Resulting single cell libraries of cDNA are prepared for HiSeq 2500 (Illumina) RNA sequencing. **(B)** Single cell transcriptomes of Raji B cells separate from control Caco-2 and co-culture cells along the first axis (PC1) of a principal component analysis projection (Raji B $n$ = 8, Caco-2 $n$ = 5, co-culture cells $n$ = 50). **(C)** Single cell transcriptomes of co-culture cells and control Caco-2 cells, visualized by principal component analysis, suggesting the progressive acquisition of an M cell phenotype in co-culture Caco-2 cells (Caco-2 $n$ = 5, co-culture cells $n$ = 50). **(D)** Subsets of single co-culture cells express higher levels of genes of the RANKL / RANK M cell induction pathway, compared to control Caco-2 cells. Subsets of single co-culture cells express higher levels of genes of the epithelial-mesenchymal transition (EMT) pathway, compared to control Caco-2 cells (Caco-2 $n$ = 5, co-culture cells $n$ = 50).
(TIF)

**S4 Fig. Scheme of the workflow for live imaging of *in vitro* M cell infections. (A)** Caco-2 co-culture M cells (magenta) and enterocytes (beige) are cultured on the membrane of a transwell. Stable fluorescent reporters and dyes are used to identify subcellular bacterial localizations, bacteria (red), label M cells and distinguish cellular membranes live. Apical initiation of bacterial infection is performed in an upright configuration to allow bacterial deposition on the epithelium by gravity. **(B)** Upon apical interaction of the bacteria with the epithelium, the transwell membrane is excised and adhered upside-down to the base of an optical dish with the apical side of the epithelium facing the bottom of the dish. **(C)** Optical infection medium is added to the dish and the sample is acquired up to 21 hours by time-lapse imaging using an inverted confocal microscope.
(TIF)

**S5 Fig. Preserved surface composition and functionality during live imaging procedures in co-cultures.** (**A**) A similar fold-change increase of GP2 positive M cells is observed in co-cultures excised and glued upside down for 1 hour, compared to non treated co-cultures, versus non treated monocultures ($n$ = 3). (**B**) A similar fold-change increase of WGA positive cells is observed in co-cultures excised and glued upside down for 1 hour, compared to non treated co-cultures, versus non treated monocultures ($n$ = 3). (**C**) Superglue treatment of co-cultures does not affect *S. flexneri* transcytosis ($n$ = 3).
(TIF)

**S6 Fig.** Complementation assays for *S. flexneri* ΔicsA and *L. monocytogenes ΔactA phenotypes* (A) IcsA complementation rescues the ability of *S. flexneri ΔicsA* to spread and form large infection niches at late time-points. The statistical significance of differences between conditions and WT were assessed by one sample t tests, and differences between conditions were assessed by a two-tailed unpaired t test ($n$ = 3) (**$p < 0.01$, ***$p < 0.001$). (B) ActA complementation rescues the ability of *L. monocytogenes ΔactA* to subvert transcytosis through the co-cultures ($n$ = 3) (**$p < 0.01$). (C) ActA complementation partially rescues the ability of *L. monocytogenes ΔactA* to spread and form large infection niches at late time-points, discussed in S1 Appendix. The statistical significance of differences between conditions and WT were assessed by one sample t tests, and differences between conditions were assessed by a two-tailed unpaired t test, ($n$ = 3)(*$p < 0.05$, **$p < 0.01$, ***$p < 0.001$).
(TIF)

**S7 Fig.** *L. monocytogenes* **virulence effectors LLO, pIcA and pIcB do not play a role in subversion of M cell transcytosis.** (**A**) The LLO mutant *L. monocytogenes* Δ*actA* does not transcytose through M cell containing co-cultures (*n* = 4). (**B**) The triple mutant *L. monocytogenes* Δ*hly* Δ*pIcA* Δ*pIcB* does not transcytose through M cell containing co-cultures (*n* = 5).
(TIF)

**S1 Movie. Related to** Fig 2 *S. flexneri* **gains access to the M cell cytosol upon vacuolar rupture.** Representative time-lapse of vacuolar rupture and intracellular replication of *S. flexneri* within a previously identified M cell, visible through galectin-3-eGFP recruitment (arrow) and binary fission of dsRed bacteria. Time-lapse confocal images were taken every 5 min for 7 hours. Data from three independent experiments. Scale bar, 5 μm.
(MOV)

**S2 Movie. Related to** Fig 3 *S. flexneri* **spreads directly from the M cell cytosol to neighboring cells.** Representative time-lapse of the promotion of actin-based motility and intercellular spread by *S. flexneri* within a previously identified M cell, visible by the formation of villin-eYFP positive comet tails at the pole of dsRed bacteria (asterisk) and the appearance of FM4-64 positive membrane protrusions (arrow) into the neighboring cell. Time-lapse confocal images were taken every 5 min for 7 hours. Data from three independent experiments. Scale bar, 5 μm.
(MOV)

**S1 Appendix. Description of immunofluorescence-, transcytosis- and single-cell transcriptomic-based characterization of the human** *in vitro* **FAE model.**
(DOCX)

## Acknowledgments

We thank Felix A. Rey and John Rohde for critical comments of the manuscript. We thank Philippe Sansonetti, Javier Pizarro-Cerda and Pascale Cossart for bacterial strains. We thank Marie-Agnes Dillies for advice on single cell transcriptomic statistical analysis, Valentina Libri and the Center for Translational Science (CRT) / Cytometry and Biomarkers Unit of Technology and Service (CB UTechS) at Institut Pasteur for support in conducting this study; and Jean-Yves Tinevez and the Imagopole France–BioImaging infrastructure, supported by the French National Research Agency (ANR 10-INSB-04-01, Investments for the Future), for advice and access to microscopes. Finally, we thank Gianfranco Grompone and Muriel Derrien for enabling this study.

## Author Contributions

**Conceptualization:** Camille Rey, Jost Enninga.

**Data curation:** Hugo Varet, Rachel Legendre.

**Formal analysis:** Camille Rey, Hugo Varet.

**Funding acquisition:** Camille Rey, Jost Enninga.

**Investigation:** Camille Rey, Yuen-Yan Chang, Patricia Latour-Lambert, Caroline Proux.

**Methodology:** Camille Rey, Patricia Latour-Lambert.

**Project administration:** Jean-Yves Coppée, Jost Enninga.

**Resources:** Jean-Yves Coppée, Jost Enninga.

**Software:** Hugo Varet.

**Supervision:** Jean-Yves Coppée, Jost Enninga.

**Validation:** Camille Rey, Hugo Varet.

**Visualization:** Camille Rey, Hugo Varet.

**Writing – original draft:** Camille Rey, Jost Enninga.

**Writing – review & editing:** Camille Rey, Yuen-Yan Chang, Jost Enninga.

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
