## [Decision Letter · Decision Letter 0]

4 Dec 2019

Dear Dr Enninga,

Thank you very much for submitting your manuscript "Shigella flexneri and Listeria monocytogenessubvert microfold cell immunological sampling by direct actin-based M cell-to-Enterocyte dissemination" (PPATHOGENS-D-19-01874) for review by PLOS Pathogens. Your manuscript was fully evaluated at the editorial level and by independent peer reviewers. The reviewers appreciated the attention to an important problem, but raised some substantial concerns about the manuscript as it currently stands. These issues must be addressed before we would be willing to consider a revised version of your study. We cannot, of course, promise publication at that time.

We therefore ask you to modify the manuscript according to the review recommendations before we can consider your manuscript for acceptance. Your revisions should address the specific points made by each reviewer. Please pay particular attention to the following reviewer comments/suggestions in your revised manuscript:

Relevance of the model. The physiological relevance of Shigella entry into M cells is controversial. There is little evidence of M-cell colonization by Shigella in humans or higher primates, where the primary site of colonization is the colon, which is largely devoid of M-cells. Rather the “widely accepted” M-cell colonization model has largely been derived from ligated ileal loop studies in rabbits, mice and guinea pigs. The authors should present a more balanced view of the literature in the Discussion and also tone down some of their “sweeping conclusions”. This applies not only to the text, but also the manuscript title.Complementation of Shigella mutants (Major point 1, reviewer 2).Statistical analysis (Reviewer 3)The validity of gluing monolayers before live cell imaging (Reviewer 3)

(1) A letter containing a detailed list of your responses to the review comments and a description of the changes you have made in the manuscript. Please note while forming your response, if your article is accepted, you may have the opportunity to make the peer review history publicly available. The record will include editor decision letters (with reviews) and your responses to reviewer comments. If eligible, we will contact you to opt in or out.

(2) Two versions of the manuscript: one with either highlights or tracked changes denoting where the text has been changed; the other a clean version (uploaded as the manuscript file).

Additionally, to enhance the reproducibility of your results, PLOS recommends that you deposit your laboratory protocols in protocols.io, where a protocol can be assigned its own identifier (DOI) such that it can be cited independently in the future. For instructions see http://journals.plos.org/plospathogens/s/submission-guidelines#loc-materials-and-methods

We hope to receive your revised manuscript within 60 days. If you anticipate any delay in its return, we ask that you let us know the expected resubmission date by replying to this email. Revised manuscripts received beyond 60 days may require evaluation and peer review similar to that applied to newly submitted manuscripts.

Sincerely,

Leigh Knodler

Guest Editor

PLOS Pathogens

Renée Tsolis

Section Editor

PLOS Pathogens

Kasturi Haldar

Editor-in-Chief

PLOS Pathogens

orcid.org/0000-0001-5065-158X

Grant McFadden

Editor-in-Chief

PLOS Pathogens

orcid.org/0000-0002-2556-3526

Reviewer's Responses to Questions

**Part I - Summary**

Reviewer #1: In this paper, Eninga and colleagues designed a tissue culture systems to investigate the role of M cells in Shigella flexneri infection and dissemination. In this model, S. flexneri develops a bimodal lifestyle within M cells that involves (1) rapid transcytosis or (2) vacuole escape followed by actin motility-based motility and spread to adjacent enterocytes. The authors argue that their results provide a potential mechanism for the intracellular pathogen ability to evade adaptive immunity. Although the experiments are appropriately conducted, it is unclear why the authors launched such a research program to begin with. The model of S. flexneri infection involving M cells originated from studies conducted in the small intestine, which is not the site of S. flexneri infection in humans. Moreover, recent studies strongly suggest that in the colon, which lacks M cells, Shigella directly infect epithelial cells. So this reviewer appreciate the technical tour de force displayed here, but unfortunately the biological significance of the project and the approach is groundless.

Reviewer #2: M cells have been demonstrated in the past to represent a major entry port of important human enteropathogens such as Shigella and Listeria. In addition, lateral cell-to-cell spread in a monolayer of epithelial cells has been demonstrated for both Shigella and Listeria. The rapid and efficient translocation of bacteria by M cells, however, prompted the idea, that M cells provide the port of entry and that both bacterial pathogens subsequently re-enter the epithelial layer from the basolateral side to spread cell-to-cell.

Instead, Rey et al. now show that M cells provide both, (i) port of entry to allow the translocation and (ii) the initial focus of lateral epithelial spread. They demonstrate that a fraction of intracellular bacteria in M cells lyse the trafficking endosomal compartment, acquire cytosolic motility spread from the M cell to the neighboring enterocytes generating a focus of infected cells.

Although the mechanisms involved are not novel per se, the careful time-lapse resolution and differentiation between basolateral re-infection and M-cell-derived lateral cell-to-cell spread generate a significant conceptual advancement and may explain previous immunological findings. The study is well-performed using state of the art tools (reporter constructs, high end microscopy, life imaging) and is certainly of interest for a broader readership in the field of infection research and mucosal immunology. The reviewer feels that the results of the study do perfectly justify publication in PLoS Pathogens.

Reviewer #3: This manuscript by Rey et al. investigates the mechanisms of initial infection of follicle-associate epithelia (FAE) by invasive bacterial pathogens using an in vitro microfold (M) cell model. The authors describe the validation of the previously established human M cell model and determine its appropriateness to model bacterial infection. Using a combination of transcytosis measurements and confocal time-lapse microscopy, infecting Shigella flexneri are described as having two fates: either being rapidly transcytosed or lysing the M cell vacuole in a delayed manner. Use of different bacterial strains uncovers a role for actin-based motility in the lateral dissemination into neighboring enterocytes by the latter population. Furthermore, similar experiments with Listeria monocytogenes lead the authors to propose that this evasion of M cell transcytosis by actin-based motility phenomenon is not specific to Shigella.

The results generally support the overall conclusions made by the authors. However, the authors tend to make sweeping conclusions in places where the data are lacking; although the results here are likely to support the general conclusions, the story would be more convincing if the authors were more circumspect in their interpretation of the data in these places.

**Part II – Major Issues: Key Experiments Required for Acceptance**

Reviewer #1: NA

Reviewer #2: Major points :

1. If possible also provide the results of complemented bacterial strains (Fig. 2b, Fig. 3b, Fig. 4a).

2. Are there any preferences in the evasion-promoting activity of the M cell endosomal compartment as compared to the endosomal compartment of absorptive enterocytes, e.g. in respect of the lytic effectors of Listeria (Hly, PlcA or PlcB)

Reviewer #3: (No Response)

**Part III – Minor Issues: Editorial and Data Presentation Modifications**

Reviewer #1: NA

Reviewer #2: Minor points:

1. Add space: Line 87.

2. Check italic style for in vitro/in vivo and mutans.

3. Line 155, the infection time starts from 40 min.

4. "Fig 4C,D" in line 198 should be moved to line 196.

5. Line 331, µ not Italic.

6. Line 345, 5µm instead of 5um.

7. Line 493, add bracket.

Reviewer #3: 1. Live cell imaging: the technique employed was to allow bacteria to enter cells for 30 minutes, then to glue the monolayer in an inverted orientation to a mount and image live. More controls are needed to test whether critical M cell functions are not altered by using super glue to adhere the apical surface to a fixed polymer surface. Does transcytosis of beads (and S. flexneri) continue to occur at the rates that it occurs in non-inverted/glued co-cultured cells? Do the M cells continue to produce WGA ligand(s) and GP2?

2. The proposition that delay for T3SS-dependent hijacking of vacuole trafficking in M cells due to the inherent high levels of endomembrane compartment trafficking in these cells is interesting. The authors could test the impact of bacterial factors on the rate of transcytosis in M cells by doing the bead transcytosis experiments in concert with infection with bacteria wild type strains and their respective actin-based motility defective mutants. If IcsA and ActA are involved in the dysregulation of transcytosis, the number of beads recovered from co-infection with the WT strains ought to be decreased compared with those recovered from co-infection with the icsA and actA strains.

3. Fig 1: The figure legend should provide more of the relevant details. Panels B and D, define which cells are in “monoculture” and indicate that they are polarized. Panel C, define what the “control” is.

4. Fig. 2. In panel C, are all images from the same field? If so, why does the shape of the outline change at 80 and 110 minutes?

5. How were the perimeters of the individual cells (the dashed lines) determined?

6. Define WGA, GP2, UEA1, SI.

7. Lines 116-118: Although the authors avoided some of the tools that investigators use to enhance infection, it is not true that they avoided all of them, as according to the Methods, the infecting strain appears to produce the E. coli Afa-I adhesin; this should be mentioned here and in the 3rd paragraph of the Discussion (lines 238-240).

8. Lines 132-133 (Fig. 1E): “The time-lapses confirmed the exclusive apical targeting of M cells by S. flexneri…” The data presented in Fig. 1E show a representative image of S. flexneri infecting an M cell. They do not show any quantification that might support the conclusion that infection was (1) “exclusively apical” or (2) “exclusively of M cells.” Either the authors need to be more circumspect in their conclusions or show the data that support these conclusions.

9. Statistical methods: The authors state (lines 611-613) that differences between experimental conditions were determined by t tests. Whereas t tests are appropriate for experiments that consist of only two experimental conditions, for experiments in which three or more conditions were used, analysis of the data by ANOVA is more appropriate and should be used here. In addition, where two conditions are present, if the data are transformed (e.g., to percentages), the t test is also generally not appropriate. Moreover, where the t test is appropriate, the assumption of a Gaussian distribution appears to have not been assessed.

10. In Fig. 4D, the presentation of the data is statistically incorrect: since the sum of the two bars is necessarily 100%, the two bars should not be directly compared by statistics. An appropriate presentation of the data would be to have a single bar that totaled 100%, and within that bar to indicate the proportion infected cells that are enterocytes and the proportion that are M cells. Moreover, these data are really only relevant if compared to a quantification of the percentage of the cells in the analyzed monolayers that were M cells (and the percentage that were enterocytes); these numbers are not clearly presented anywhere in the manuscript.

11. Regarding L. monocytogenes, the suggestion in the Figure 4 legend title and in the Results sub-heading is that it behaves similarly to S. flexneri, whereas the results indicate that it behaves differently, in that transcytosis is detectable only in the absence of actin-based motility. These headings are therefore a bit misleading; the authors should consider rewording them to better reflect the findings.

12. As currently written, the Discussion is largely a regurgitation of the results. Therefore, it could be shortened substantially. On the other hand, how the antigen-sampling function of M cells might relate (or not) to its use by pathogens would be a good topic on which to expand the Discussion.

13. Whereas the authors claim that the system is physiological, Fig. S1C indicates that the co-culture leads to nearly 100% of cells producing the M cell markers WGA ligand(s) and GP2 and approximately 50% of cells producing the epithelial cell markers UEA1 and SI. There are two issues here that should be addressed: (1) in the human colon, M cells are the minority of cells (nowhere near 100% of cells), and (2) what type of cells are the approximately 50% in this system that produce both M cell markers and epithelial cell markers?

14. Fig. S3: Transcriptional data show that compared with monocultured cells, subsets of co-cultured cells express higher levels of genes of the RANKL/RANK M cell pathway and the epithelial-mesenchymal transition pathway. With cluster or other analysis, is it the same subset of cells that are expressing higher levels of each of the transcripts that are shown here as a separate graph or are these distinct subsets?

15. In Appendix S1, the authors state that the transcriptomics analysis of the co-cultures offers “new insights into the involvement of relevant biological pathways for M cell differentiation”. It seems to this reviewer that it is the reverse, that the transcriptomes were analyzed to assess whether expression profiles characteristic of M cells were present in the co-culture model, and thus few new insights were offered?

16. In certain places, the wording is unclear: lines 153-159 and 489-493.

17. Fig 4C: It would be informative to add an image of infection with the actA mutant.

18. References: reference 29 appears to be incorrect, with little mention of Listeria in the context of translocation across the FAE. Several references have author duplications (e.g. references 20, 26, …).

PLOS authors have the option to publish the peer review history of their article (what does this mean?). If published, this will include your full peer review and any attached files.

Reviewer #1: No

Reviewer #2: Yes: Mathias Hornef

Reviewer #3: No

---

## [Editor Report · Decision Letter 1]

29 Feb 2020

Dear Jost,

We are pleased to inform you that your manuscript 'Transcytosis subversion by M cell-to-enterocyte spread promotes Shigella flexneri and Listeria monocytogenesintracellular bacterial dissemination' has been provisionally accepted for publication in PLOS Pathogens.

Best regards,

Leigh Knodler

Guest Editor

PLOS Pathogens

Renée Tsolis

Section Editor

PLOS Pathogens

Kasturi Haldar

Editor-in-Chief

PLOS Pathogens

orcid.org/0000-0001-5065-158X

Michael Malim

Editor-in-Chief

PLOS Pathogens

orcid.org/0000-0002-7699-2064
---

## [Editor Report · Acceptance letter]

3 Apr 2020

Dear Dr. Enninga,

We are delighted to inform you that your manuscript, "Transcytosis subversion by M cell-to-enterocyte spread promotes *Shigella flexneri* and *Listeria monocytogenes*  intracellular bacterial dissemination," has been formally accepted for publication in PLOS Pathogens.

Best regards,

Kasturi Haldar

Editor-in-Chief

PLOS Pathogens

orcid.org/0000-0001-5065-158X

Michael Malim

Editor-in-Chief

PLOS Pathogens

orcid.org/0000-0002-7699-2064